# Decoding Chemotherapy Resistance of Undifferentiated Pleomorphic Sarcoma at the Single Cell Resolution: A Case Report

**DOI:** 10.3390/jcm13237176

**Published:** 2024-11-26

**Authors:** Timur I. Fetisov, Maxim E. Menyailo, Alexander V. Ikonnikov, Anna A. Khozyainova, Anastasia A. Tararykova, Elena E. Kopantseva, Anastasia A. Korobeynikova, Maria A. Senchenko, Ustinia A. Bokova, Kirill I. Kirsanov, Marianna G. Yakubovskaya, Evgeny V. Denisov

**Affiliations:** 1Research Institute of Molecular and Cellular Medicine, Peoples’ Friendship University of Russia (RUDN University), 115093 Moscow, Russia; timkatryam@yandex.ru (T.I.F.); max89me@yandex.ru (M.E.M.); alex.v.ikonnikov@gmail.com (A.V.I.); muhteshemka@gmail.com (A.A.K.); anastasiatararykova@gmail.com (A.A.T.); y.kopantseva@gmail.com (E.E.K.); shegolmay@gmail.com (A.A.K.); senchenko.mariia@yandex.ru (M.A.S.); pushkay@yandex.ru (U.A.B.); kkirsanov85@yandex.ru (K.I.K.); mgyakubovskaya@mail.ru (M.G.Y.); 2N.N. Blokhin National Medical Research Center of Oncology, 115478 Moscow, Russia; 3Cancer Research Institute, Tomsk National Research Medical Center, Russian Academy of Sciences, 634009 Tomsk, Russia

**Keywords:** chemotherapy, resistance, undifferentiated pleomorphic sarcoma, soft tissue sarcoma, recurrence, single cell RNA sequencing, bioinformatics, transcriptome

## Abstract

**Background:** Undifferentiated pleomorphic sarcoma (UPS) is a highly malignant mesenchymal tumor that ranks as one of the most common types of soft tissue sarcoma. Even though chemotherapy increases the 5-year survival rate in UPS, high tumor heterogeneity frequently leads to chemotherapy resistance and consequently to recurrences. In this study, we characterized the cell composition and the transcriptional profile of UPS with resistance to chemotherapy at the single cell resolution. **Methods:** A 58-year-old woman was diagnosed with a 13.6 × 9.3 × 6.0 cm multi-nodular tumor with heterogeneous cysto-solid structure at the level of the distal metadiaphysis of the left thigh during magnetic resonance tomography. Morphological and immunohistochemical analysis led to the diagnosis of high-grade (G3) UPS. Neoadjuvant chemotherapy, surgery (negative resection margins), and adjuvant chemotherapy were conducted, but tumor recurrence developed. The UPS sample was used to perform single-cell RNA sequencing by chromium-fixed RNA profiling. **Results:** Four subpopulations of tumor cells and seven subpopulations of tumor microenvironment (TME) have been identified in UPS. The expression of chemoresistance genes has been detected, including *KLF4* (doxorubicin and ifosfamide), *ULK1*, *LUM*, *GPNMB*, and *CAVIN1* (doxorubicin), and *AHNAK2* (gemcitabine) in tumor cells and *ETS1* (gemcitabine) in TME. **Conclusions:** This study provides the first description of the single-cell transcriptome of UPS with resistance to two lines of chemotherapy, showcasing the gene expression in subpopulations of tumor cells and TME, which may be potential markers for personalized cancer therapy.

## 1. Introduction

Undifferentiated pleomorphic sarcoma (UPS) is a group of highly malignant mesenchymal tumors with undetermined differentiation patterns. UPS is severely understudied and remains the diagnosis of exclusion [1]. UPS constitutes 17% of STS cases, ranking as the second most common STS after leiomyosarcoma and having a higher prevalence rate in males over females and in white males over black males. The incidence rate of UPS also increases with age [2]. It is hypothesized that the tumor cells of origin for UPS are mesenchymal stem cells, which causes the high heterogeneity of the tumor and the wide spectrum of its location: predominantly the extremities, followed by the trunk, the retroperitoneum, and the left atrium [3,4].

The 5-year overall survival rate for UPS is 50–70%, which is slightly lower than the average survival for other types of STS [5]. The rate of recurrence for UPS is suggested to be greater than 30% [6], with both local recurrences and distant metastases being common [7]. Chemotherapy significantly increases the 5-year survival rate in UPS, especially in tumors greater than 5 cm in size [8]. However, high tumor heterogeneity frequently leads to chemotherapy resistance and consequently to recurrences, indicating the need to reveal mechanisms of chemoresistance and to identify corresponding markers [9].

Single-cell RNA sequencing (scRNA-seq) is a highly effective method for deciphering cellular heterogeneity, allowing to identify types and subpopulations of tumor and tumor microenvironment (TME) cells and to analyze individual cell transcriptional profiles. Previous scRNA-seq studies of UPS showed several specific transcripts enriched in tumor and TME cells [10,11,12]. Tumor cells probably originate from mesenchymal stem cells [11] and non-tumor-propagating cells [10], which are capable of self-renewal [10]. Recently, UPS chemoresistance has been investigated using drug testing in vivo [13] and in vitro [14], real-time PCR [15,16], and bulk RNA sequencing [17,18]. However, its mechanisms and specific markers remain unclear.

Here, we first used scRNA-seq to characterize a case of UPS that was resistant to first-line (doxorubicin and ifosfamide) and second-line (docetaxel and gemcitabine) chemotherapy. As a result, potential genes (*KLF4*, *ULK1*, *ETS1*, *AHNAK2*, *LUM*, *GPNMB*, and *CAVIN1*) were identified to be associated with resistance to chemotherapy.

## 2. Case Report

We present the clinical case of undifferentiated pleomorphic soft tissue sarcoma, which was localized on the left thigh of a 58-year-old female patient. The patient’s history is structured in Figure 1A.

In January 2023, the patient sought medical attention due to a significant increase in the size of the tissue formation and the additional painful sensation. The magnetic resonance tomography of the thigh joint was performed, which revealed the presence of the 13.6 × 9.3 × 6.0 cm multi-nodular tumor with heterogeneous cysto-solid structure and vague uneven contours, located in the soft tissues of the posterior lateral surface at the level of the distal metadiaphysis of the left thigh (Figure 2A,B). The patient was redirected to the N.N. Blokhin National Medical Research Center of Oncology, where, under the control of ultrasound navigation with the use of a semi-automatic G14 needle, the core biopsy of the formation was performed. Five columns of tissue were collected and sent to morphological analysis. Soft tissue tumor was represented by fields and accumulations of atypical explicitly polymorphic and spindle-like cells, multi-lobe, locally multinucleated cells, and abundant cytoplasm (Figure 2C,D). In order to clarify the diagnosis, the immunohistochemical analysis was conducted with antibodies to S-100, CD31, SOX-10, ERG, CDK4, caldesmon, SMA, Myo-D1, and MyoG. The results demonstrated diffuse expression of caldesmon, the focal expression of ERG and CDK4, and the absence of expression of S-100, CD31, SOX-10, SMA, Myo-D1, and MyoG in tumor cells. According to the immunohistochemical scores, the final diagnosis of the high grade (G3) UPS (Fédération Nationale des Centres de Lutte Contre Le Cancer, FNCLCC) was established.

This case was discussed during the multidisciplinary reference consilium, which decided that given the size, localization, extent of spreading (stage IIIB, T3N0M0) and the histological type of the tumor, the neoadjuvant chemotherapy will be conducted according to the standard scheme: doxorubicin 60 mg/m^2^ once in a day, ifosfamide 2500 mg/m^2^ once in 1–3 days, and Mesna (2-mercaptoethane sulfonate Na) 3000 mg/m^2^ once in 1–3 days, with a 21-day break period between courses. Magnetic resonance and computer tomography of the lower limbs showed the multi-nodular tumor growth to 15 × 7.6 × 6.8 cm. A few small blood vessels were visualized inside the tumor architecture. During the intermediate follow-up, an increase in the volume of the cystic component and the appearance of diffuse fibrotic areas were noted inside the tumor structure. The pre-surgery control check revealed an increase in the volume of areas of ossification inside the tumor.

In July 2023, the surgical removal of the tumor was performed. One tumor sample was used for histological analysis, another one—for scRNA-seq. The tumor consisted of conglomerates of atypical explicitly polymorphic and spindle-like cells with multi-lobe and, at times, multi-nuclear nuclei and abundant cytoplasm. Some tumor regions had pronounced hyalinosis, small areas of necrosis, and stromal edema. The percentage of retained viable tumor cells was 65%. The resection margins were negative (R0).

Considering a large amount of intact viable tumor cells, the patient was once again discussed at the multidisciplinary consilium, during which the adjuvant chemotherapy administration has been changed. The treatment scheme included six therapy courses with a mid-treatment checkpoint after the third course and a concluding checkpoint after the sixth final course. Taking into account the radical extent of the surgical operation, the risks of development of distant metastasis (in addition to local recurrence), the histological type of the tumor, and its size, the recommendations were to conduct adjuvant polychemotherapy according to the scheme, gemcitabine and docetaxel, instead of radiotherapy. In September 2023, three courses of adjuvant chemotherapy were carried out: gemcitabine 900 mg/m^2^ on days 1 and 8, and docetaxel 100 mg/m^2^ on day 8, with a time interval of 21 days. After 3 courses (November 2023), there were no signs of local relapse or distant progression. Between 12 December 2023 and 22 February 2024, three more courses of adjuvant chemotherapy were carried out according to the previous scheme.

Four months after the treatment, a multi-nodular recurrent tumor was detected in soft tissues of the left thigh along the contours of fluid accumulation in the area of post-operational changes in the close proximity of pelvic blood vessels and the thigh. Considering the localization and the prevalence rate of tumor recurrence, surgical removal of the lower limb was suggested, which was turned down by the patient. The alternative treatment was isolated limb perfusion for stabilizing the tumor growth. On 15 August 2024, the isolated limb perfusion was performed on the patient’s left lower limb. At present, the patient is being dynamically monitored.

The regimen of choice for the first line of therapy of localized non-operable and/or metastatic soft tissue sarcomas are the schemes containing anthracyclines. The choice of therapy takes into account the localization of the process, the patient’s age, the previous history of treatment, the possibility of surgical treatment in the presence of the oligometastatic state, and the morphological tumor type. Combined chemotherapy regimens (doxorubicin and ifosfamide, doxorubicin and dacarbazine, gemcitabine and docetaxel) successfully increase the tumor progression-free survival and the frequency of the objective response in comparison with monotherapy. In the case of disease progression, the scheme of choice for the second line of treatment, given the overall satisfactory status of the patient, is the combination of gemcitabine with docetaxel. The decision to proceed with the neoadjuvant and adjuvant chemotherapy in the case of localized soft tissue sarcomas has to be made by the multidisciplinary consilium at the reference sarcoma centers. The key indicators are the highly malignant and sensitive to chemotherapy sarcoma type, the high risk of metastasis and recurrence, and the size and localization of the tumor. Pre-surgery chemotherapy increases the probability of the surgical stage being conducted to the extent of R0 and increases the survival rate prior to the progression and the patient’s quality of life. During treatment pathomorphosis of grade 3 and higher (after neoadjuvant therapy), it is generally recommended to conduct 2–4 courses according to the previous scheme. In the case of treatment pathomorphosis of lesser grade than 3 or R1, it is recommended to conduct radiotherapy in combination with the second line of therapy or without it. In the absence of pre-surgery therapy for high-risk sarcomas, it is recommended to conduct adjuvant therapy, which includes radiotherapy and/or chemotherapy [19,20,21].

Radiotherapy combined with surgery is necessary in the case of high risk of recurrence to improve the local control. For localized soft tissue sarcomas of the extremities and trunk, neoadjuvant radiotherapy is preferable in the presence of indicators for radiotherapy. The surgical operation on the first stage with adjuvant radiotherapy is preferable during a complicated disease progression (pain, tumor disintegration, bleeding) or during the high risk of post-operational complications: tumor greater than 10 cm in size, the proximity of the tumor to skin, tumor localized in the lower extremities, smoking, diabetes, accompanying diseases of the blood vessels, obesity. Adjuvant radiotherapy starts 4–6 weeks after surgery and under the condition that the post-operation wound is healed. In the case of an unresectable tumor or the patient not giving consent to the operation, it is possible to conduct radiotherapy independently. The target volumes and doses correspond to neoadjuvant radiotherapy with an additional boost (single or consecutive doses) to the final dose of 63 Gy. The treatment scheme and the order in which the procedures are conducted are decided by a multidisciplinary team, which includes a surgeon, a chemotherapist, a morphologist, and a radiotherapist. The parameters of the tumor, its localization, morphological subtype and grade, the comorbidity status of the patient, the possibility of radical removal without heavy functional and cosmetic defects, the risks of post-operation complications, and the possibility of repeat surgery in the case of a recurrence are all taken into account [19,20,21].

The surgical tumor sample was fixed by the Chromium Single Cell Fixed RNA Sample Preparation Kit (10× Genomics, Inc., Pleasanton, CA., USA). Cell suspension was prepared by the Tumor Dissociation Kit (Miltenyi Biotec, Cologne, Germany) on the DSC-410 Single Cell Suspension Dissociator (RWD, Shenzhen, China). Cell counting was carried out using an acridine orange/propidium iodide stain buffer (Logos Biosystems, Anyang, Republic of Korea). ScRNA-seq libraries were prepared using the Chromium Fixed RNA protocol (10× Genomics, Inc., Pleasanton, Cal., USA) and sequenced on the Genolab M platform (GeneMind, Shenzhen, China) by 28 cycles for read 1 and 90 cycles for read 2. The scRNA-seq data was demultiplexed and aligned to the GRCh38-2020-A reference genome using the Cell Ranger 7.1.0 (10× Genomics, Inc., Pleasanton, CA, USA) pipeline. In total, 2275 cells with a median of 3885 reads and 1302 genes per cell have been identified. A total of 17,102 genes were detected across the sample, and the median UMI count was 1953 per cell. The total number of reads was 19,952,047, including 80.86% of them mapped to cells, indicating high data quality. Bioinformatic analysis was conducted using the Seurat package 5.0.3 [22]. The preprocessing quality control (QC) workflow included the following parameters: 500 < nFeature < 6000, 500 < nCount < 20,000, percent mt < 5%, and genes present in at least three cells. The estimated doublet rate was 1.6%, and doublet detection was performed using the DoubletCollection package v. 1.1.0 [23]. Further analysis included 1922 cells that remained after filtering using quality metrics. Data normalization was carried out using the SCTransform method applied to the expression matrices. The remaining cells were grouped into 11 clusters using the Leiden algorithm from the Seurat package v. 5.0.3 (Figure 1B). Differentially expressed genes (DEGs) were identified using Seurat with the MAST algorithm [24]. Statistical thresholds for DEGs were set at a log fold change > 1 and an adjusted *p*-value < 0.05, corrected for multiple testing using the Benjamini–Hochberg method, and a min.pct parameter of 0.5. Following the aneuploidy analysis using the SCEVAN package v. 1.0.1 [25], 5 cell clusters (1, 2, 4, 6, and 10) were singled out as containing the highest number of aneuploid cells (32–53%; 465 cells in total), considered to be tumor cells (Figure 1C). Four clusters of aneuploid cells were identified after reclusterization (Figure 1D). The majority of tumor cells were found in clusters 1, 2, and 3 (33%, 26%, and 30% of cells, respectively), while cluster 4 contained 11% of total cells. Cell type-specific markers identified among DEGs were used to annotate cell clusters. Additionally, tumor cells were annotated through a combination of aneuploidy analysis and DEG profiling after reclustering. Enrichment analysis was conducted on Kyoto Encyclopedia of Genes and Genomes (KEGG) signaling pathways and Gene Ontology (GO) 2023 terms (biological processes, molecular functions, and cellular components) using the EnrichR package v. 3.2 [26]. This comprehensive approach facilitated a detailed characterization of each cell type, enhancing the interpretation of functional differences across clusters.

Based on DEGs analysis, the identified tumor cell subpopulations were characterized as PCDH1^+^ (*PCDH1*, *STRA6*, *SEC14L2*, *C5AR2*, and *KLF4*; cluster 1), PLEKHG5^+^ (*PLEKHG5*, *TNFRSF25*, *TNNT3*, *ECM2*, and *THBS3*; cluster 2), LUM^+^ (*LUM*, *C1R*, *GPNMB*, *COL1A1*, and *PRSS23*; cluster 3), and IQGAP3^+^ cells (*IQGAP3*, *TROAP*, *DLGAP5*, *AURKB*, and *KIF20A*; cluster 4) (Figure 3A,B, Table 1 and Appendix A). PCDH1^+^ tumor cells harbored the *RHOB*, *C5AR2*, *HAPLN3*, *SERPINE2*, *ELN*, *EPS8L2*, and *ITGA5* genes, associated with cell migration and metastasis [27,28,29] (Figure 3C and Appendix A). PLEKHG5^+^ tumor cells had the *COL16A1*, *ECM2*, *FAM118A*, *LSP1*, *TNFRSF25*, *TNNT3*, *PLPP1*, and *MEIS1* genes associated with immune cell infiltration (Figure 3C, Supplement Appendix A). LUM^+^ tumor cells were mainly enriched in proteoglycans in cancer and extracellular matrix organization and demonstrated expression of the *MMP2*, *MMP14*, *COL1A1*, *COL6A2*, and *COL1A2* genes, associated with extracellular matrix remodeling (Figure 3C and Figure 4A, Appendix A). IQGAP3^+^ tumor cells were predominantly enriched in cell cycle and mitosis, indicating their proliferating potential (Figure 5A). The TOP10 genes and the most significant signaling pathways and biological processes are given in Figure 3C and Figure 4A; the complete lists are shown in Appendix A.

Seven cell subpopulations were identified in TME according to cell-specific markers: macrophages (*CD163*, *CD74*, *MS4A7*, *MS4A6A*, and *PSAP*; cluster 3), T cells (*CD2*, *CD96*, *CYTIP*, *TRAC*, and *TMSB4X*; cluster 5), endothelial cells (*PLVAP*, *HSPG2*, *VWF*, *PECAM1*, and *CLEC14A*; cluster 7), COL4A1^+^ fibroblasts (*COL18A1*, *NOTCH3*, *COL4A2*, *ACTA2*, and *COL4A1*; cluster 8), COL11A1^+^ fibroblasts (*POSTN*, *COL11A1*, *CDH11*, *DCN*, and *ASPN*; cluster 9), lymphatic endothelial cells (*FLT4*, *PROX1*, and *CCL21*; cluster 11), and mast cells (*CPA3*, *MS4A2*, and *HDC*; part of cluster 5) (Figure 4A,B, Table 2 and Appendix A). Macrophages contained markers of M2 differentiation (*MS4A7*, *C1Q*, *CD163*, and *VSIG4*) [30,31,32,33] and the *FOLR2* gene expression associated with interactions between macrophages and T cells [34] (Figure 4C, Appendix A). T cells harbored the *ETS1* and *IL2RG* genes, associated with cell growth and metastasis [35,36] (Figure 5B and Appendix A). Endothelial cells were mainly enriched in regulation of angiogenesis, cell migration, and blood vessel morphogenesis (Figure 5B). COL4A1^+^ fibroblasts had the high expression of *NOTCH3*, *COL18A1*, *MCAM*, and *ACTA2* genes and enrichment in vascular smooth muscle contraction and artery development, which allowed us to identify them as vascular cancer-associated fibroblasts (vCAF) [37] (Figure 4C and Figure 5B and Appendix A). COL11A1^+^ fibroblasts demonstrated the high expression of collagens (*COL11A1*, *COL14A1*, *COL12A1*, *COL1A1*, *COL3A1*, *COL6A2*, *COL6A1*, and *COL6A3*) and metalloproteases (*MMP2* and *ADAMTSL1*) genes and were enriched in negative regulation of angiogenesis, which suggests an antagonistic relationship between these two fibroblast subpopulations (Figure 4B,C and Appendix A). Lymphatic endothelial cells were predominantly enriched in angiogenesis and positive regulation of cell migration (Figure 5B). Mast cells showed enrichment in the Fc epsilon RI signaling pathway, cytokine-mediated signaling pathway, and prostaglandin metabolic process, indicating their activation in immune response and inflammation (Figure 5B). These cells also exhibited the high expression of *PTGS1*, *CPA3*, and *HDC* genes, which are closely associated with inflammatory response and allergic reaction [38] (Figure 4C). The TOP10 genes and the most significant signaling pathways and biological processes are given in Figure 4C and Figure 5B; the complete lists are shown in Appendix A.

## 3. Discussion

Despite the fact that surgical resection is the main treatment method for UPS, chemotherapy plays a vital preventive role on cancer recurrence and metastasis. Nevertheless, UPS is often characterized by chemotherapy resistance, which leads to disease progression. The rarity and high heterogeneity of UPS make revealing chemoresistance mechanisms and predictive markers difficult. In this study, we used scRNA-seq to conduct the first analysis of tumor and TME landscape in the UPS patient with resistance to the first (doxorubicin and ifosfamide) and second (docetaxel and gemcitabine) lines of chemotherapy.

The tumor subpopulations identified by scRNA-seq have not been previously identified for UPS, which might be explained by the high heterogeneity of this disease. PCDH1^+^ tumor cells had a high expression of *PCDH1* and *ITPR3* genes previously described as associated with metastasis [39,40] and *KLF4*, *ULK1*, and *AHNAK2* genes, which are related to resistance to doxorubicin, ifosfamide, and gemcitabine [41,42,43,44]. *KLF4* was shown to contribute to chemoresistance in osteosarcoma partially through regulation of the high-mobility group box 1 (HMGB1) [41]. Furthermore, treatment of osteosarcoma cells with doxorubicin activates KLF4, which promotes stemness and metastatic potential of cancer cells [42]. The AMPK-ULK1 signaling pathway triggers protective autophagy in doxorubicin-resistant breast-cancer cells upon treatment with doxorubicin [43]. *AHNAK2* is expressed at high levels in gemcitabine-resistant pancreatic cancer cells, where it promotes proliferation through activating the KRAS/p53 signaling pathway and impacts cell adhesion and TME [44]. LUM^+^ tumor cells demonstrated a high expression of *LUM*, *GPNMB*, and *CAVIN1* genes linked to low sensitivity to doxorubicin [45,46,47]. *LUM* expression was detected in cytostatic-resistant ovarian cancer cell lines, where this protein was implicated in the collagen fiber assembly [45]. In chondrosarcoma cells, LUM and IGF-IR interact and consequently activate the AKT signaling pathway, which positively regulates cell proliferation and inhibits apoptosis [48]. Interestingly, activation of the AKT signaling pathway is observed in the LUM^+^ tumor cells. Therefore, the effects of LUM and AKT interaction described previously by Papoutsidakis et al. may be observed in the present UPS case, which provides chemoresistance to the given cell subpopulation through inhibition of cell death pathways. GPNMB promotes cancer cell stemness and metastasis, and its expression is triggered by standard chemotherapy agents in triple-negative breast cancer [46]. PTRF/Cavin-1 promotes multidrug resistance in MCF-7/ADR cancer cell lines through fortification of lipid rafts, which enhances cancer signal transduction [47]. Moreover, as shown above, PLEKNG5^+^ has been identified in UPS, which supposedly interacts with the immune system. For instance, a high expression of the *LSP1* gene has been observed in the PLEKNG5^+^ subpopulation. Previously, it has been shown that the *LSP1* knockdown in the murine B16 melanoma leads to infiltration of the tumor with T cells and a decrease in the rate of tumor growth [49]. Additionally, the same tumor subpopulation demonstrated a high expression of the *ECM2* gene. A high expression of ECM2 has been connected to glioma infiltration with immune cells and a short overall survival rate, which suggests a possible mechanism of immune cell recruitment by tumor cells [50].

In the UPS TME, T cells expressed the *ETS1* and *IL2RG* genes associated with resistance to gemcitabine [51] and tumor metastasis [36], respectively. ETS1 expression was detected in gemcitabine-resistant pancreatic cancer cells, and its silencing leads to the partial reversal of gemcitabine chemoresistance [51]. UPS TME was predominantly represented by M2 macrophages. Previous reports demonstrated that M2 macrophages are associated with progression and resistance to various chemotherapeutics, including docetaxel [52], gemcitabine [53], and doxorubicin [54]. Therefore, M2 macrophages may be considered a promising target for UPS therapy. Additionally, M2 macrophages were characterized by heightened lysosomal activity, including acidity, and the high expression of proteases. The use of the pH-gated nanoadjuvant, which selectively interacts with the lysosomes of M2 macrophages, leads to macrophage reprogramming into the M1 pro-inflammatory/anti-tumor state, the activation of the immune response, and the consequent reduction of the tumor size [55].

In the last 20–30 years, the median overall survival rate for patients with UPS has not changed. The effectiveness of therapeutic approaches is often limited due to the development of resistance to treatment. The genes described above can be used for personalized prescription of chemotherapeutics and increase in UPS treatment efficacy. In addition, these genes may be potential targets for developing new therapies for UPS.

The study has several limitations. The results have been obtained on only one case and should be validated in independent samples. In addition, UPS was analyzed only at one time point after neoadjuvant chemotherapy, whereas scRNA-seq of UPS before chemotherapy and of recurrence after a second line of chemotherapy is needed to get a better understanding of chemoresistance mechanisms. Nevertheless, the current transcriptional profile contains both markers of resistance to the first line of chemotherapy and potential markers of resistance to the second line.

## 4. Conclusions

UPS is a group of highly malignant mesenchymal tumors with poor prognosis that are often resistant to chemo- and radiotherapy. In this study, we used single-cell RNA sequencing to identify potential mechanisms and markers involved in UPS chemoresistance. UPS was found to have high expression of the *KLF4*, *ULK1*, *LUM*, *GPNMB*, *CAVIN1*, *AHNAK2*, and *ETS1* genes, which are associated with insensitivity to doxorubicin, ifosfamide, and gemcitabine according to previous studies. These genes can be predictive markers for personalized treatment and potential targets for novel therapies in UPS.

## Figures and Tables

**Figure 1 jcm-13-07176-f001:**
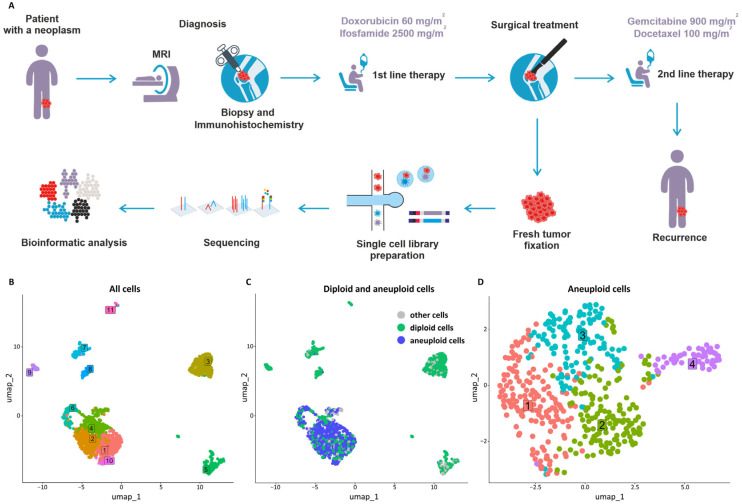
ScRNA-seq of UPS. (**A**) The patient’s history and experimental workflow. (**B**) UMAP plot of all cells. (**C**) UMAP plot of aneuploid and diploid cells. (**D**) UMAP plot of aneuploid cells. 1–11, cluster numbers.

**Figure 2 jcm-13-07176-f002:**
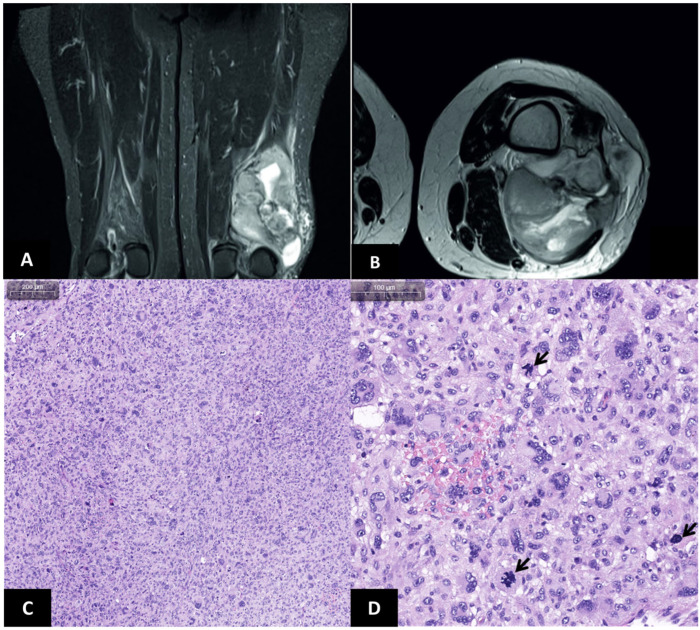
Magnetic resonance and histological imaging of UPS. (**A**) Thigh before treatment, frontal section. (**B**) Thigh before treatment, axial section. (**C**) Hypercellular neoplastic tissue with prominent cellular pleomorphism. (**D**) Frequent bizarre multinucleated cells and high mitotic activity (arrows).

**Figure 3 jcm-13-07176-f003:**
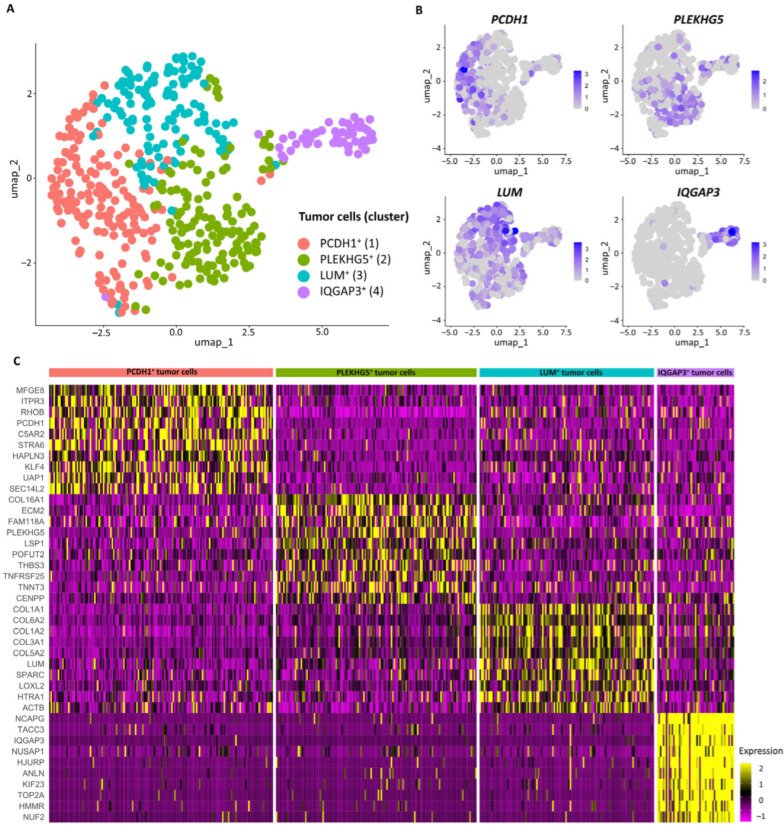
The transcriptional landscape of UPS tumor cells. (**A**) UMAP plot of tumor cell subpopulations. Clusters were annotated using aneuploidy analysis, marker genes, and functional annotation. (**B**) The gene UMAP plots showing the expression levels of tumor cell subpopulation-specific markers. (**C**) Heatmap of TOP 10 differentially expressed genes in tumor cell subpopulations.

**Figure 4 jcm-13-07176-f004:**
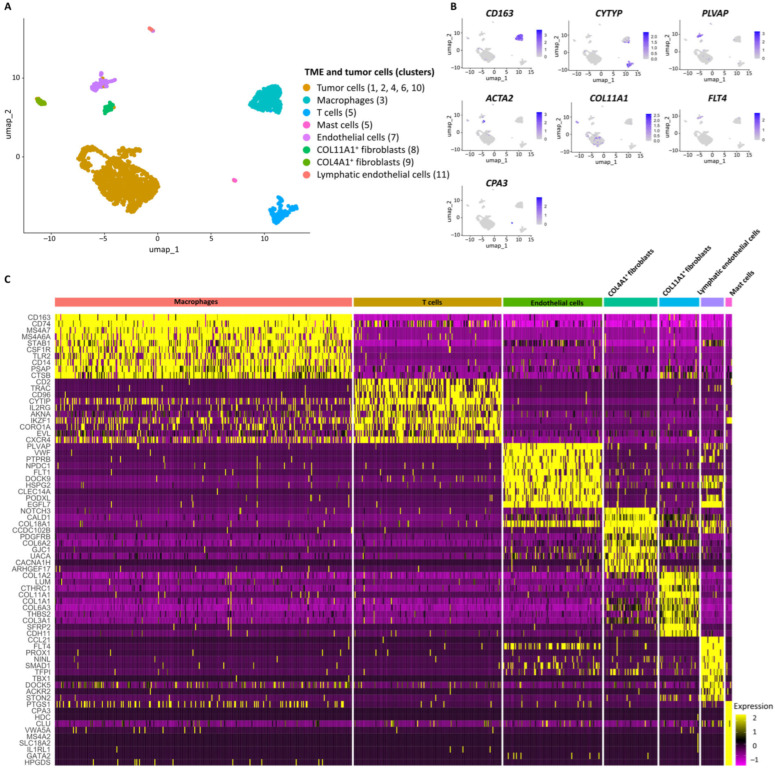
The transcriptional landscape of UPS tumor microenvironment (TME). (**A**) UMAP plot of TME cell types and tumor cell subpopulations. Clusters were annotated using marker genes and functional annotation. (**B**) The gene UMAP plots showing the expression levels of TME cell subpopulations-specific markers. (**C**) Heatmap of TOP 10 differentially expressed genes in TME cell types and tumor cell subpopulations.

**Figure 5 jcm-13-07176-f005:**
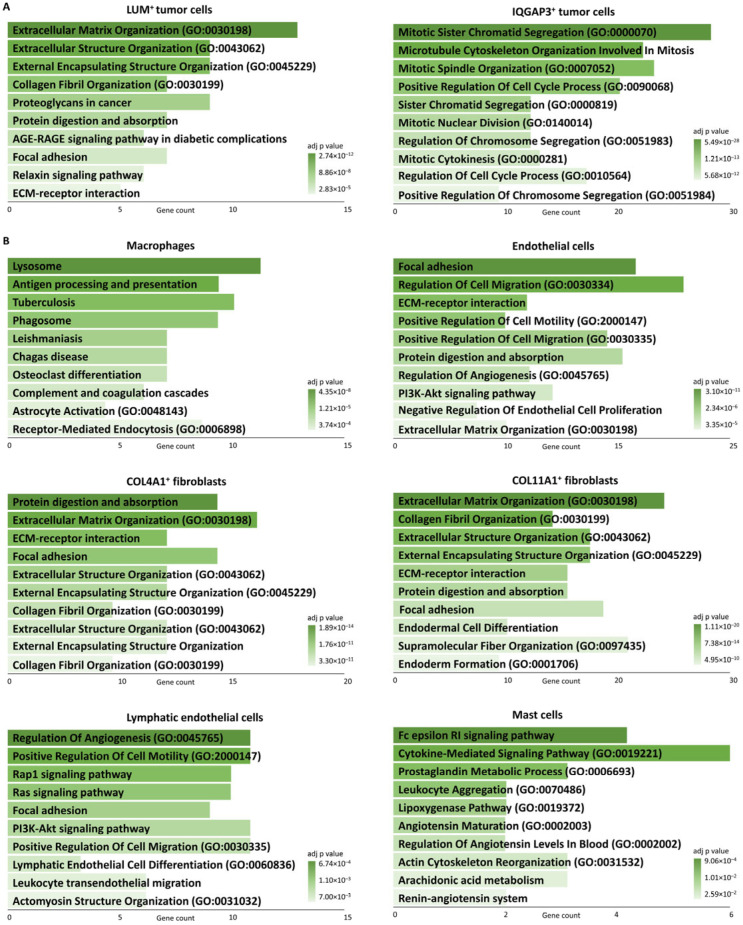
KEGG signaling pathways and GO terms enriched in UPS. (**A**) Tumor cell subpopulations. (**B**) TME cell types and subpopulations. PCDH1^+^ tumor cells had only one GO term: Integrated stress response signaling (GO:0140467). PLEKHG5^+^ tumor cells had no GO and KEGG terms.

**Table 1 jcm-13-07176-t001:** Key molecular features of UPS tumor cell subpopulations.

Subpopulations	Marker Genes	Functionally Enriched Terms
PCDH1^+^	*PCDH1*, *STRA6*, *SEC14L2*, *C5AR2*, *KLF4*	TNF signaling pathway, Extracellular matrix-receptor interaction, Integrated stress response
PLEKHG5^+^	*PLEKHG5*, *TNFRSF25*, *TNNT3*, *ECM2*, *THBS3*	Pluripotency of stem cells, Circadian rhythm
LUM^+^	*LUM*, *C1R*, *GPNMB*, *COL1A1*, *PRSS23*	Proteoglycans in cancer, Protein digestion and absorption, Extracellular matrix organization
IQGAP3^+^	*IQGAP3*, *TROAP*, *DLGAP5*, *AURKB*, *KIF20A*	Cell cycle, Mitotic sister chromatid segregation

**Table 2 jcm-13-07176-t002:** Key molecular features of UPS TME subpopulations.

Subpopulations	Marker Genes	Functionally Enriched Terms
Macrophages	*CD163*, *CD74*, *MS4A7*, *MS4A6A*, *PSAP*	Antigen processing and presentation, phagosome, inflammatory response
T cells	*CD2*, *CD96*, *CYTIP*, *TRAC*, *TMSB4X*	Regulation of actin cytoskeleton, negative regulation of NF-kappaB transcription factor activity, antigen processing and presentation of peptide antigen via MHC Class I, regulation of lymphocyte differentiation
Endothelial cells	*PLVAP*, *HSPG2*, *VWF*, *PECAM1*, *CLEC14A*	Extracellular matrix-receptor interaction, focal adhesion, Regulation of angiogenesis, regulation of cell migration
COL4A1^+^ Fibroblasts	*COL18A1*, *NOTCH3*, *COL4A2*, *ACTA2*, *COL4A1*	Extracellular matrix organization, basement membrane organization
COL11A1^+^ Fibroblasts	*POSTN*, *COL11A1*, *CDH11*, *DCN*, *ASPN*	Protein digestion and absorption, collagen fibril organization
Lymphatic endothelial cells	*FLT4*, *PROX1*, *CCL21*	PI3K-Akt signaling pathway, regulation of endothelial cell proliferation, Lymphatic endothelial cell differentiation
Mast cells	*CPA3*, *MS4A2*, *HDC*	Mast cell mediated immunity

## Data Availability

The data are available from the corresponding author upon reasonable request.

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
