# Peer review of "Decoding Chemotherapy Resistance of Undifferentiated Pleomorphic Sarcoma at the Single Cell Resolution: A Case Report"

_jcm, 2024, doi:10.3390/jcm13237176_

Round 1

Reviewer 1 Report

Comments and Suggestions for Authors

The authors presented a case of undifferentiated pleomorphic sarcoma that exhibited recurrence and resistance to standard chemotherapy regimens. They also aimed to understand the mechanisms underlying the aggressive behavior of this tumor using single-cell RNA sequencing. The idea is intriguing, the methods are appropriate and adequate, and the manuscript is well-written.

Some points:

1-       The authors have used the term “multi-node tumor” on several occasions. I think “multi-nodular tumor” is a better choice.

2-        Lines 63 and 115: I think “hip” should be replaced by “knee”.

3-       “Multinuclear nuclei” is better to be replaced by “multinucleated cells”.

4-       “Calcidesmon” is probably a typo. I guess the authors meant “Caldesmon”.

5-       Line 114: “Multimodal” is probably a typo.

6-       It is recommended to enrich the discussion by including more details from the referenced sources.

Reviewer 2 Report

Comments and Suggestions for Authors

This case report by Fetisov et al. provides an insightful and pioneering analysis of a case of UPS resistant to two chemotherapy lines using scRNA-seq. The study is of high relevance, as it addresses the challenges posed by chemoresistance in UPS, a rare and highly heterogeneous malignancy. The authors’ approach to profiling tumor and ùTME subpopulations offers valuable insights into potential genetic markers of resistance and lays the groundwork for personalized therapeutic strategies.

Comment:

- While the introduction effectively covers the complexity and treatment challenges of UPS, adding context on its prevalence, clinical significance, and common outcomes compared to other sarcomas would enhance the study's relevance for readers less familiar with this tumor type.

- The choice of specific genes for chemoresistance analysis, such as KLF4, ULK1, and AHNAK2, would benefit from a brief rationale. Explaining their roles in prior studies on UPS or similar cancers would enhance understanding of their significance in this case.

- methodology is clear but could be improved by specifying the criteria for cell inclusion and exclusion in preprocessing (e.g., expression range, doublet removal). Additionally, where available, include metrics on RNA sample quality to reinforce the robustness of the scRNA-seq data.

- Although Seurat and clustering algorithms are appropriate, a concise explanation of the process used to annotate cell subpopulations (both tumor and TME) would improve transparency, especially for readers less familiar with bioinformatics methods.

- The four identified tumor subpopulations are well-presented; however, it remains somewhat unclear how functional differences, particularly between PLEKHG5+ and LUM+ cells, relate directly to chemoresistance or metastasis. An expanded description of these implications would clarify the clinical relevance of each subtype.

- The expression of KLF4, ULK1, LUM, and other markers associated with resistance is interesting, but the functional context remains somewhat underexplored. Including a brief discussion on how each gene might contribute to resistance pathways in UPS or similar cancers would solidify their role in this context.

- The authors discuss cellular heterogeneity and resistance effectively. However, expanding on how these findings might inform future therapeutic strategies would add value. Specific suggestions on potential drug targets aligned with the gene profiles (e.g., cycle inhibitors for IQGAP3+ or M2-targeted therapies for TME macrophages) would underscore the study's translational impact.

- Given the novel molecular profile presented, adding a brief reflection on the clinical feasibility of using these scRNA-seq findings in treatment personalization, along with current limitations, would provide a realistic view of both challenges and possibilities.

- The UMAP plots of cell clusters are informative but would benefit from a more detailed legend explaining cluster characteristics. This would enhance accessibility for readers who may be unfamiliar with scRNA-seq visualization.

- The supplementary tables are comprehensive, yet a main text summary table listing key marker genes for each cell subgroup (with brief functional annotations) could streamline data interpretation and highlight key findings.

Reviewer 3 Report

Comments and Suggestions for Authors

This is a report of single-cell analysis using a resected specimen of undifferentiated pleomorphic sarcoma. It is well-described and has a certain academic significance. This paper requires the following revisions.

#1. This case appears to be of soft tissue origin, not bone origin. However, the description is misleading, as it suggests that it is of femoral origin. The word “femur” should be changed to “thigh”.

#2. The clinical and histological images of this case are not shown at all. In order to increase the reliability of this report, it is essential to add these images.

#3. The description of m2 in Figure 1 should be changed to m2.

#4. It appears that the authors are suggesting that the profiles of the tumor cells at the time of initial diagnosis and those at the time of recurrence are the same. This is not guaranteed, and this report's results should only be related to resistance to 1st-line treatment.

Comments on the Quality of English Language

Minor English editing is necessary 

Reviewer 4 Report

Comments and Suggestions for Authors

The article presents a case report of a 58-year-old woman with undifferentiated pleomorphic sarcoma (UPS) resistant to two lines of chemotherapy. Single-cell RNA sequencing (scRNA-seq) was employed to analyze the tumor and its microenvironment, identifying gene expression patterns associated with chemotherapy resistance. The study provides new insights into the tumor's heterogeneity and highlights potential markers for personalized therapy.

The study successfully identifies genes associated with resistance to multiple chemotherapy agents, which can be useful for future therapeutic strategies.

The findings offer valuable markers for developing personalized treatments for UPS . patients, addressing a critical gap in managing this aggressive cancer.

Despite that this study is case report , limiting the generalizability of the findings to other UPS cases. It still provide an interesting results for futures studies

some area to be modified:

add a paragraph of common chemotherapy used (first line , second line etc)

The role of chemotherapy in non metastasis and in metastasis context (this should be well detailed as chemotherapy has no rule in certain sarcomas, in case of adjuvant or neoadjuvant)

add a small paragraph regarding radiotherapy

Add site , depth of the sarcoma and margins in the abstract

add MRI image if possible , exact location in regard with NV structures

provide exact margins ( limited margins)

why radiotherapy was not done ? , this should will be explained in the case report and in the discussion.

increase the conclusion
decrease similarity index (40% is too much and need to be less than 15%)

Round 2

Reviewer 2 Report

Comments and Suggestions for Authors

reviewers have responded to the comments.